# CD34+ Hematopoietic Stem Cell Counts in Alzheimer’s Disease: A Meta-Analysis

**DOI:** 10.3390/diseases13020025

**Published:** 2025-01-21

**Authors:** Vinay Suresh, Malavika Rudrakumar, Anmol Kaur, Victor Ghosh, Poorvikha Satish, Amogh Verma, Priyanka Roy, Mainak Bardhan

**Affiliations:** 1King George’s Medical University, Lucknow 226003, India; 2St. John’s Medical College, Bengaluru 560034, India; 3Lady Hardinge Medical College, New Delhi 110001, India; 4Andhra Medical College, Visakhapatnam 530002, India; 5Rama Medical College Hospital and Research Centre, Hapur 245304, India; 6Chief Inspector of Factories, Deputy Director (Medical) and Certifying Surgeon, Directorate of Factories, Department of Labour, Government of West Bengal, India; 7Miller School of Medicine, University of Miami, Miami, FL 33136, USA

**Keywords:** Alzheimer’s disease (AD), CD34+ cells, endothelial progenitor cells, vascular biomarkers, meta-analysis

## Abstract

Purpose: To assess the presence and quantity of CD34+ hematopoietic stem cells in patients with Alzheimer’s disease (AD) through a meta-analysis. Methods: A systematic search of the databases identified the observational and interventional studies reporting baseline CD34+ cell counts in AD patients. The data on mean counts and the measures of variation were extracted. Standardized mean differences (SMDs) were calculated using common and random effects models to compare the CD34+ cell counts between the AD patients and controls. Heterogeneity among the studies was evaluated using tau^2^, tau, and I^2^ statistics. The risk of bias was assessed using the Newcastle–Ottawa Scale and the ROBINS-I tool. Patients: Five studies were included, comprising four observational studies and one open-label trial, with a total of 271 participants (139 AD patients and 132 controls). Results: The meta-analysis indicated an increase in CD34+ cell counts of the AD patients when compared to the controls. The common effects model showed a moderate SMD of 0.2964 (95% CI:0.0490–0.5437). However, the random effects model yielded a non-significant SMD of 0.2326 (95% CI: −0.4832–0.9484). Significant heterogeneity was observed among the studies (I^2^ = 87.1%, *p* < 0.0001). Conclusion: AD patients may exhibit higher circulating CD34+ cell counts than the controls, but substantial heterogeneity and potential biases limit definitive conclusions.

## 1. Introduction

Alzheimer’s disease (AD) is a slow, progressive neurodegenerative disorder characterized by dementia and cognitive impairment [1]. Current estimates suggest that approximately 50 million individuals worldwide are affected by this condition [2]. While the precise etiology of AD remains challenging to decipher, it is often linked to age-related degeneration. It has been hypothesized that alterations in brain vasculature may be an underlying driver of many of the pathophysiological mechanisms of AD. Key factors such as dysfunctional angiogenesis and compromised blood–brain barrier integrity are believed to play critical roles in the onset and progression of AD. Therefore, it is pertinent to understand the vascular and angiogenic modifications associated with AD.

Endothelial progenitor cells, which possess the characteristics of both endothelial and stem cells, play a crucial role in angiogenesis and endothelial maintenance. They express specific markers such as CD34, CD133, KDR, and/or CD146. Multiple studies have shown that CD34+ progenitor cells demonstrate stage-dependent upregulation, suggesting potential vascular repair processes in the brain [3]. It has also been postulated that CD34+ cells have a role in neuro-regenerative processes including differentiation between neural cells and neovascularization [4]. The quantity of circulating progenitor cells is thus theorized to be a reliable indicator of cerebrovascular function and repair capacity and has been observed to decline with advancing age [3].

In this review, we aim to differentially evaluate the presence and number of CD34+ cells in individuals afflicted with Alzheimer’s disease. By delving into this complex interplay, insights into the pathological processes underlying this neurodegenerative condition can be gained, potentially facilitating the development of novel therapeutic interventions.

## 2. Methodology

### 2.1. Literature Search and Screening

We conducted a systematic literature search across multiple databases, including MEDLINE (PubMed), EMBASE, Cochrane, Scopus, and clinicaltrials.gov. The inclusion criteria required the studies involve populations diagnosed with Alzheimer’s disease and to have analyzed CD34+ cell counts. We also included interventional studies that reported baseline cell counts prior to the initiation of any intervention. Studies were excluded if they had ineligible study designs like case reports and editorials. Animal studies were excluded. This study did not register.

The search query used is detailed in Appendix A. All identified studies were imported into Rayyan.ai for screening [5]. Two independent reviewers assessed the studies, and any conflicts were resolved by a third reviewer or through a discussion with the team. The PRISMA guidelines were followed for reporting the review process, and the flowchart was generated using the tool developed by Haddaway et al. (Appendix A) [6].

### 2.2. Data Extraction and Analysis

CD34+ cell count data for the Alzheimer’s patients and control groups were extracted and recorded in a spreadsheet, including mean counts and the measures of variation for each group. Two primary analyses were performed using R. The first analysis compared the mean CD34+ cell counts between the Alzheimer’s patients and controls, utilizing the standardized mean difference (SMD) and Hedges’ g to circumvent the heterogeneity introduced by the differing units across studies. The SMD was reported with its 95% confidence intervals. Multi-group studies were combined to a single pairwise comparison based on Cochrane guidelines. The second analysis focused on the mean CD34+ cell counts within the Alzheimer’s group, using the Inverse Variance method, with the mean and its 95% confidence interval reported. Additionally, the studies were qualitatively analyzed and reported to provide a more comprehensive inference.

### 2.3. Risk of Bias Assessment

The Risk of Bias was evaluated using the Newcastle–Ottawa Scale (NOS) for observational studies (Appendix A) [7]. The assessment was conducted and reported in accordance with the standards set by the Agency for Healthcare Research and Quality (AHRQ). The non-randomized interventional studies were assessed for a risk of bias using the ROBINS-I tool (Appendix A) [8].

## 3. Results

A total of 733 articles were initially identified through our search across multiple databases. After de-duplication, 227 articles were removed, leaving 506 unique records. Of these, 30 articles met the criteria for full-text evaluation. Reports were excluded for the following reasons: no results posted (*n* = 3), reviews (*n* = 3), post-mortem tissue-based studies (*n* = 4), results deemed irrelevant to the research question (*n* = 7), irrelevant markers (*n* = 7), and ineligible outcome measures (*n* = 1). Ultimately, five studies were deemed eligible for inclusion in this meta-analysis (Figure 1).

A total of four studies with 271 observations were included in the analysis (Table 1), comprising 139 observations from the Alzheimer’s group and 132 from the control group. The analysis revealed an SMD of 0.2964 (95% CI: 0.0490–0.5437) under the common effect model and the random effects model yielded an SMD of 0.2326 (95% CI: −0.4832–0.9484) (Figure 2). The heterogeneity analysis indicated substantial variability among the studies, with tau^2 estimated at 0.4680 (95% CI: 0.1038–7.5195) and tau at 0.6841 (95% CI: 0.3222–2.7422). The I^2 statistic was 87.1% (95% CI: 69.0% to 94.6%), suggesting high heterogeneity. The test for heterogeneity was significant (Q = 23.26, df = 3, *p* < 0.0001).

**Figure 2 diseases-13-00025-f002:**
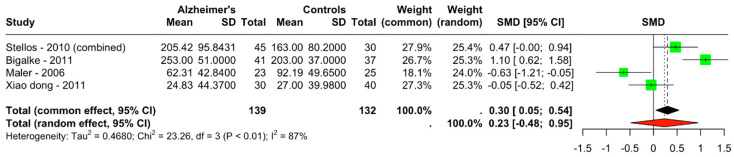
Forest plot of standardized mean differences (SMDs) in CD34+ cell counts between Alzheimer’s disease patients and controls [3,4,9,10].

**Table 1 diseases-13-00025-t001:** Population characteristics of the studies included in the meta-analysis.

Authors and Year	Study Design	Population	Age	Sex	Outcome	Conclusion
Kong et al. (2011). [9]	Case–Control Study	Patients with newly diagnosed AD (*n* = 30), vascular dementia (*n* = 34), and healthy control with normal cognition (*n* = 40).	71.4 ± 2.3	15:15	CD34+ and CD133+ levels.	Endothelial function is impaired in patients with AD. Patients with AD have reduced circulating endothelial progenitor cells, CD 34+ cells, and CD 113+ cells.
Stellos et al. (2010). [4]	Case–Control Study	Patients with previously diagnosed AD (*n* = 45) who fulfilled the ICD-10, DSM-IV, and NINCDS-ADRDA criteria and healthy controls (*n* = 30).	73.2 ± 9.3	27:18	MSME and CDR were assessed. Peripheral blood mononuclear cell isolation and flow cytometry, and ELISA were completed.	The number of CD34+/CD133+ or CD34+ progenitor cells is increased in AD patients.
Bigalke et al. (2011). [3]	Case–Control Study	Patients diagnosed with early AD (*n* = 41) who fulfilled the ICD-10, DSM-IV, and NINCDS-ADRDA criteria and healthy controls (*n* = 37).	74.3 ± 9.1	22:19	MSME was assessed. Flow cytometry and ELISA were performed to measure levels of CD34+ cells, Leptin, and Adiponectin.	Circulating CD32+ cell count significantly upregulated in AD patients. Plasma level of Leptin was found to be significantly reduced in AD patients.
Lewis et al. (2013). [11]	Interventional Study	Patients with moderate to severe AD (*n* = 34).	79.9 ± 8.4	28:6	Neuropsychological battery, ADAS-Cog, MSME, and ADCS-ADL were assessed. Multiplex Cytokine and Growth factor levels were measured. T cell subsets and levels of CD14, CD34, CD90, and CD95 subsets were measured.	A relative and absolute decrease was noted in the lymphocyte region (CD90+, CD95+CD3+, CD95+CD34+, and CD95+CD90+) from baseline.
Maler et al. (2006). [10]	Case–Control Study	Patients with newly diagnosed, CSF-based neurochemically confirmed early AD (*n* = 25) and age–sex-matched healthy controls (*n* = 25).	-	-	CD4+, CD8+, CD3+/CD56+, CD3+/CD25+, CD4+/CD25+, CD4+/CD28+, CD8+/CD25+, CD45+/CD133+, and CSF A(beta)1–42 levels were measured.	Hematopoietic stem cell count (CD34) decreased significantly, correlating with age in early AD.

A total of three studies were included in the analysis of the mean CD34+ cell counts [3,4,11]. The common effect model reported a mean count of 241.8387 (95% CI: 227.7885–255.8888), whereas the random effects model reported a mean count of 279.1495 (95% CI: 136.6796–421.6193) (Figure 3). The heterogeneity analysis showed substantial variability among the studies, with tau^2 estimated at 14,536.6946 (95% CI: 2479.8164 → 145,366.9462) and tau at 120.5682 (95% CI: 49.7978 → 381.2702). The I^2 statistic was 92.3% (95% CI: 80.6–96.9%), indicating very high heterogeneity. The test for heterogeneity was significant (Q = 25.83, df = 2, *p* < 0.0001).

We assessed the risk of bias for the included studies using the appropriate tools tailored to the studies’ designs. The four case–control trials were evaluated using the Newcastle–Ottawa Scale and were all determined to have a low risk of bias, with minimal concerns about selection, comparability, and exposure criteria. However, the open-label pilot trial, assessed using the ROBINS-I tool, was determined to hold a serious risk of bias.

## 4. Discussion

This systematic review aimed to evaluate the presence and number of CD34+ cells in individuals with Alzheimer’s disease (AD) to better understand the role of vascular factors in the pathogenesis of neurodegenerative conditions (Figure 4).

Four observational studies were identified for inclusion in this review [3,4,9,10]. The results of our study indicate an increase in the circulating CD34+ cells in the AD group compared to the controls. Using the common effects model, there was a moderate standardized mean difference (SMD) between the AD and control groups. However, when accounting for between-study variability, the random effects model yielded a non-significant effect size. This implies that any true difference in the CD34+ cell counts may be obscured by the substantial heterogeneity among the studies. Possible reasons for the heterogeneity across the studies can be attributed to clinical differences in methodology such as variations in the diagnostic criteria used for enrolling patients with AD, the demographics of the participants who are diverse in age, gender, ethnicity, stage of AD, as well as comorbidities, and publication bias such as selective reporting which may lead to a skewed perception of the diagnostic efficacy of CD34+ as a biomarker [12].

The existing literature on the association between CD34+ cell counts and AD is inconsistent, with some studies reporting an increase and others a decrease in these cells among AD patients. Of the studies included in this review, the findings of Stellos et al. [4]. and Bigalke et al. [3] demonstrated that the number of CD34+ cells was significantly upregulated in AD patients as opposed to Kong et al. [9], Lewis et al. [11], and Maler et al. [10] who found that AD patients had decreased levels of circulating CD34+ cells in comparison to cognitively normal subjects.

Another important mechanism to note is that smoking induces the upregulation of the G-protein coupled receptor 15 (GPR15), which is known to influence regulatory T cell (Tregs) populations [13]. Tregs play a critical role in maintaining immune homeostasis, including CD34+ cells, and preventing excessive inflammation, which is a hallmark of AD pathology.

In Parkinson’s disease (PD), the neuroprotective effects of smoking have been well established, attributed to mechanisms such as reduced immune activation and the alpha-synuclein interactions mediated by HLA-DRB1, where individuals lacking protective HLA-DRB1 AA alleles exhibit a reduced PD risk [14]. The V11 residue in HLA-DRB1 has been implicated in weaker immune responses to alpha-synuclein-derived peptides due to a reduced binding affinity which may, in turn, confer neuroprotective effects by reducing inflammation [15]. However, this inverse relationship between smoking and PD cannot be translated to AD, where the beta-amyloid and tau-derived peptides are pathologically different from PD [16,17]. Although preclinical studies have explored enhancing Tregs as a therapeutic strategy for improving cognition in AD patients, ref [18], current evidence predominantly suggests that smoking increases AD risk, likely through mechanisms such as oxidative stress, vascular dysfunction, and immune dysregulation which may outweigh any neuroprotection conferred by Tregs [19]. Future studies should focus on gene–environment interactions, like those using Mendelian Randomization (MR), and Human Leucocyte Antigen (HLA) imputation, which could help elucidate smoking’s role in AD and clarify whether smoking’s effects on AD risk are mediated through causal pathways or confounding factors. Another promising avenue for future research is the role of transposable elements like SINE-VNTR-Alu (SVA) and Alu repeats in regulating gene expression, including HLA and CD34, through epigenetic modifications and transcriptional regulation which has been explored in PD [20]. Understanding such genomic and epigenomic factors can help uncover the novel mechanisms underlying AD pathogenesis.

Although the precise etiopathogenesis of AD is unknown, the accumulation of β-amyloid plaques and tau tangles are well-established characteristics of the disease. Maler et al. [10] observed that CD34+ counts were negatively correlated with the levels of β-amyloid plaques (Aβ_1–42_) and the Aβ_42/40_ ratio in cerebrospinal fluid, both of which are important biomarkers in AD pathogenesis. There is also a growing body of evidence to suggest that a reduction in CD34+ progenitor cells is directly related to vascular dysfunction leading to increased BBB permeability, which causes neurotoxicity and, ultimately, cognitive impairment [21,22]. In patients with advanced AD, Lee et al. [23] discovered that lower endothelial progenitor cell-colony forming units (EPC-CFUs) were independently associated with lower scores on the mini-mental state exam (MMSE) and higher scores on the Clinical Dementia Rating scale (CDRS), although the CD34+ counts were not distinctly different between the demented and non-demented patients. These findings are congruous with other studies where CD34+ levels were directly correlated with memory test scores [24] and CD34+ cell counts were lower in the patients with moderate to severe AD than their cognitively normal counterparts [4,9,25].

Contrarily, a study by Breining et al. [26] did not observe any significant differences in the circulating levels of CD34+ cells and the early or late EPCs between the demented and non-demented patients. Therefore, longitudinal studies are warranted to fully confirm and understand the effect of low levels of CD34+ cells on cognitive state as well as AD progression. Exploring the endothelial cells expressing the CD34+ factor may not only help with the development of adjunctive diagnostic or prognostic biomarkers for AD dementia but also facilitate the development of novel, targeted therapeutic approaches for AD in the future [27]. Clinical trials exploring the efficacy of EPC-derived exosomes or secretomes hold immense potential in AD research as they may help circumvent the adverse effects attendant with the existing cell-based therapies [28]. Therapeutic strategies targeting β-amyloid plaque deposition, such as Aducanumab and lecanemab, have shown potential in reducing amyloid plaques in the brain and mitigating cognitive decline in Alzheimer’s disease patients [29]. Additionally, natural compounds like Radix Rehmanniae have demonstrated efficacy in reducing Aβ accumulation and improving neural function in AD models [30].

Interventions that enhance synaptic plasticity and promote neural regeneration are also being actively explored to address the multifaceted pathophysiology of AD. These strategies include increasing neurotrophic expression, such as the brain-derived neurotrophic factor (BDNF), inhibiting acetylcholinesterase through agents like donepezil, and employing physiotherapy techniques like near-infrared light and transcranial magnetic stimulation [31]. By focusing on increasing neuron numbers, improving neuron survival, and enhancing synaptic activity, these approaches hold significant potential for mitigating cognitive decline and supporting neural regeneration in Alzheimer’s disease patients. Collectively, these findings emphasize the importance of therapies that target both vascular and neural components in addressing the complexity of AD.

Overall, our study suggests an increase in the CD34+ cell counts in AD patients compared to controls; however, these results must be interpreted with caution due to the substantial heterogeneity and the concerns regarding the risk of bias in the included studies. Given the limited number of eligible studies, there were no exclusions based on the risk of bias assessment. Future research should focus on controlling the potential confounding factors and standardizing the methodologies to minimize variability, thereby enhancing the comparability of studies. We recommend reporting CD34+ cell counts as absolute values per microliter to homogenize data, facilitating ease in the pooling of data for meta-analyses.

## 5. Conclusions

Our findings indicate a potential increase in CD34+ cells among AD patients when compared to controls, though the substantial heterogeneity among the studies and the associated risk of bias necessitates a cautious interpretation. By standardizing the methodologies, future research studies can enhance data comparability and clarify the role of vascular factors like CD34+ cells as potential diagnostic or prognostic biomarkers in AD patients, paving the path for novel therapeutic strategies.

## Figures and Tables

**Figure 1 diseases-13-00025-f001:**
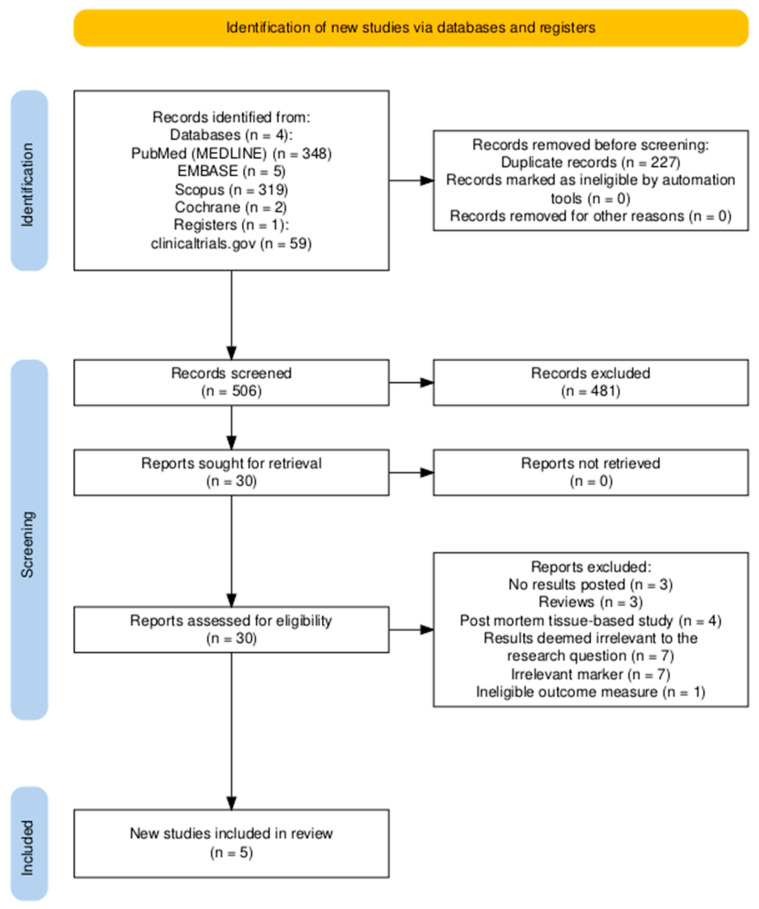
PRISMA flow diagram of study selection process in the meta-analysis of CD34+ cell counts in Alzheimer’s disease patients.

**Figure 3 diseases-13-00025-f003:**
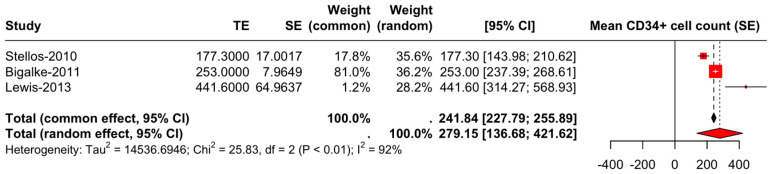
Forest plot of mean CD34+ cell counts in Alzheimer’s disease patients: random and common effect models [3,4,11].

**Figure 4 diseases-13-00025-f004:**
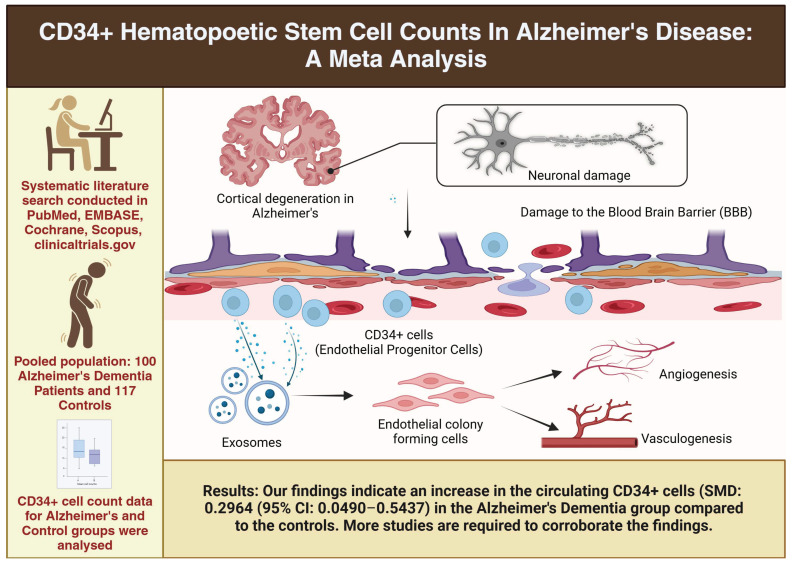
Summary of increased CD34+ hematopoietic stem cell counts in Alzheimer’s disease patients—a meta-analysis.

## Data Availability

The data supporting the findings of this study are available from the corresponding author upon reasonable request.

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
