# Peer review of "CD34+ Hematopoietic Stem Cell Counts in Alzheimer’s Disease: A Meta-Analysis"

_diseases, 2025, doi:10.3390/diseases13020025_

Round 1
Reviewer 1 Report
Comments and Suggestions for Authors
This is a lovely manuscript, but it needs additional discussion to broaden the topic.
1. CD34 is regulated by T-regs, which in turn are regulated by GPR15 (PMID: 26348578, 28423922). Smoking upregulates GPR15, which induces several health issues in smokers. Smoking also protects against Parkinson's disease (PMID: 34633332 ). What is the situation with Alzheimer's? In the case of PD, smoking interacts with HLDDRB1(PMID: 35810454).
2. HLA and CD34 are regulated by SVAs and ALu repeats, and the mechanisms have been shown here (PMID: 38590523). Do the authors have any ideas or opinions about this interaction and their findings?
These are the points that require more detailed discussion to improve this meta-analysis. These points improve the translatability of the results of the present manuscript.
Author Response
Reviewer 1
This is a lovely manuscript, but it needs additional discussion to broaden the topic.
Comments
1. CD34 is regulated by T-regs, which in turn are regulated by GPR15 (PMID: 26348578, 28423922). Smoking upregulates GPR15, which induces several health issues in smokers. Smoking also protects against Parkinson's disease (PMID: 34633332 ). What is the situation with Alzheimer's? In the case of PD, smoking interacts with HLDDRB1(PMID: 35810454).
Response:
Thank you for your suggestion. As mentioned, smoking upregulates GPR15, which can influence T-reg populations and has been associated with various health issues, including immune dysregulation. The neuroprotective effects of smoking in Parkinson's disease (PD) have been well-documented, and the interaction between smoking and HLA-DRB1 is an intriguing mechanism contributing to reduced PD risk. However, as you rightly pointed out, these findings do not directly translate to Alzheimer's disease, as the pathophysiological mechanisms, particularly the involvement of beta-amyloid and tau, differ significantly from those in PD.
In Alzheimer's, while preclinical studies have explored enhancing T-regs as a potential therapeutic strategy, current evidence predominantly suggests that smoking may increase the risk of AD. This may be due to mechanisms such as oxidative stress, vascular dysfunction, and immune dysregulation, which could outweigh any potential neuroprotective effects of T-reg modulation.
We have discussed the effect of smoking on AD progression in the Discussion ( Page 6-7, lines 147-171).
Comments
2. HLA and CD34 are regulated by SVAs and ALu repeats, and the mechanisms have been shown here (PMID: 38590523). Do the authors have any ideas or opinions about this interaction and their findings?
These are the points that require more detailed discussion to improve this meta-analysis. These points improve the translatability of the results of the present manuscript.
Response:
We appreciate this comment. Indeed, recent studies have highlighted the involvement of transposable elements, including SVAs and Alu repeats, in regulating gene expression through mechanisms such as epigenetic modifications and insertional mutagenesis . These elements could play a critical role in modulating immune responses and vascular health, both of which are relevant to Alzheimer's disease pathogenesis. Incorporating these genomic and epigenomic factors into future research could lead to the identification of novel biomarkers and therapeutic targets. We have addressed this in the Discussion of the revised manuscript ( Page 7, lines 167-171).
Reviewer 2 Report
Comments and Suggestions for Authors
This article presents a meta-analysis evaluating the presence and number of CD34+ hematopoietic stem cells in patients with Alzheimer's disease. Given that both dysfunctional angiogenesis and compromised integrity of the blood-brain barrier appear to be critical in the onset and progression of AD, CD34+ progenitor cells are potential targets for early diagnosis and treatment of the disease.
The authors showed that there was an increase in circulating CD34+ cells in the group of Alzheimer's patients compared to the control group. However, due to the heterogeneity of the studies used for analysis, age of patients, demographic differences and other reasons, the appropriateness of the described conclusions is questionable. The work is of significant interest in the context of CD34+ hematopoietic stem cell variability in the background of Alzheimer's disease. However, despite the comprehensive methodology, limited number of clinical data and findings presented, there are aspects that require further discussion.
Line 24 In the introduction of the article, it would be of interest for readers to explain the possible mechanism of variability of CD34+ Hematopoietic Stem Cell Counts in Alzheimer's Disease, including studies on animal models.
Line 118 Even if the aim of the paper was not to include experimental animal studies, assessing the correlation and reasoning about the similarity of the variability with animals would be important in the discussion of the paper.
In conclusion, this meta-analysis is important in understanding the processes of neurodegeneration and possible prognostic effects in Alzheimer's disease, but the introduction of discussions on possible correlations with animal models could increase its importance.
Author Response
Reviewer 2
Comments
This article presents a meta-analysis evaluating the presence and number of CD34+ hematopoietic stem cells in patients with Alzheimer's disease. Given that both dysfunctional angiogenesis and compromised integrity of the blood-brain barrier appear to be critical in the onset and progression of AD, CD34+ progenitor cells are potential targets for early diagnosis and treatment of the disease.
The authors showed that there was an increase in circulating CD34+ cells in the group of Alzheimer's patients compared to the control group. However, due to the heterogeneity of the studies used for analysis, age of patients, demographic differences and other reasons, the appropriateness of the described conclusions is questionable. The work is of significant interest in the context of CD34+ hematopoietic stem cell variability in the background of Alzheimer's disease. However, despite the comprehensive methodology, limited number of clinical data and findings presented, there are aspects that require further discussion.
Response:
We thank the reviewer for this comment. We acknowledge that the heterogeneity observed warrants caution when interpreting the results. We have discussed this limitation in the manuscript and highlighted the need for future studies to address these confounding variables, such as standardizing diagnostic criteria, controlling for demographic differences, and increasing the sample size to improve the robustness of the findings. (Pages 7-8 , lines 201- 206 )
Comments
Line 24 In the introduction of the article, it would be of interest for readers to explain the possible mechanism of variability of CD34+ Hematopoietic Stem Cell Counts in Alzheimer's Disease, including studies on animal models.
Response:
Thank you for this suggestion. While there are no studies that only measure the levels of CD34+ cells in animal models, there are certainly experimental models in the literature that have looked into the utility of CD34+ cells in the therapy of Alzheimer’s disease. We have included some such studies in the discussion section.
Comments
Line 118 Even if the aim of the paper was not to include experimental animal studies, assessing the correlation and reasoning about the similarity of the variability with animals would be important in the discussion of the paper.
Response:
Thank you for the suggestion. We have now included the outcomes of certain animal model studies in the discussion section.
Comments
In conclusion, this meta-analysis is important in understanding the processes of neurodegeneration and possible prognostic effects in Alzheimer's disease, but the introduction of discussions on possible correlations with animal models could increase its importance.
Response:
Thank you for the suggestion. We have included the outcomes of certain animal model studies in the discussion section, lines 198 to 207, page 7-8 of the manuscript.
Round 2
Reviewer 1 Report
Comments and Suggestions for Authors
They have addressed all my comments, and MS can now be accepted.